# Quantification of *Anopheles* daily sugar feeding rates in Siaya county, western Kenya using Attractive Sugar Baits

Jackline Jeruto Kosgei[1,2*], Daniel P. McDermott[3], Seline Omondi[1], Vincent Moshi[1], Martin J. Donnelly[3], Collins Ouma[2], Prashanth Selvaraj[4], Angela F. Harris[5], Julian Entwistle[5], Feiko O. Ter Kuile[6], Keith Fraser[7], Lazaro Mwandigha[7], Bernard Abong'o[1], John E. Gimnig[8], Eric Ochomo[1,3]

1 Entomology Section, Centre for Global Health Research, Kenya Medical Research Institute, Kisumu, Kenya, 2 Department of Biomedical Sciences and Technology, School of Public Health and Community Development, Maseno University, Maseno, Kenya, 3 Department of Vector Biology, Liverpool School of Tropical Medicine, Liverpool, United Kingdom, 4 Institute for Disease Modeling, Gates Foundation, Seattle, Washington, United States of America, 5 Innovative Vector Control Consortium, Liverpool, United Kingdom, 6 Department of Clinical Sciences, Liverpool School of Tropical Medicine, Liverpool, United Kingdom, 7 Imperial College London, London, United Kingdom, 8 Division of Parasitic Diseases and Malaria, Centers for Disease Control and Prevention, Atlanta, Georgia, United States of America

* jackieruto@yahoo.com

## Abstract

### Background

Vector control is an essential component of malaria prevention that has contributed to the reduction in malaria burden since 2000. Although steady progress in malaria vector control has been achieved over the years, the malaria burden remains substantial, underscoring the need for complementary mosquito control tools to further reduce transmission. Attractive targeted sugar baits (ATSBs) are a novel vector control tool under evaluation. The ATSB paradigm leverages the sugar feeding and resting behavior of mosquitoes exposing them to the lethal effect of an added insecticide. Prior to epidemiological trials on ATSBs in western Kenya, validation studies were conducted to assess the levels of mosquito feeding on attractive sugar baits (ASBs), containing uranine dye. This study sought to understand the ATSB deployment required in peridomestic spaces and to determine the daily feeding rates that would be potentially sufficient to impact malaria transmission (based on modelling approaches). The study evaluated whether the deployment of two versus three bait stations per structure led to higher daily feeding rates by local malaria vectors that is consistent with the modelled threshold of 2.5% of all mosquitoes in the target area as a proxy for ATSB efficacy resulting in a 30% drop in clinical incidence in children under 5.

**Data availability statement:** The data has been deposited to Zonedo and can be accessed through the following link: https://doi.org/10.5281/zenodo.17570657.

**Funding:** This study was funded by IVCC through support from the Bill and Melinda Gates Foundation (INV- 007509), the Swiss Agency for Development and Cooperation (SDC) (Grant: 81067480) and UK Aid (Grant: 30041-105). The findings and conclusions contained within are those of the authors and do not necessarily reflect positions or policies of IVCC, the Bill & Melinda Gates Foundation, SDC or UK Aid.

**Competing interests:** The authors have declared that no competing interests exist.

## Methods

The study followed a cross-over design in ten treatment and two control clusters within Rarieda Sub-County, Siaya County, western Kenya. Within each intervention cluster, either two or three ASBs were deployed to all structures in each cluster. After two months, the treatments were switched so that clusters which initially received two ASBs were given three ASBs and vice versa. ASB monitoring was done for four months during the initial crossover trial and then for an additional four months for extended monitoring. Mosquitoes were collected using ultraviolet CDC light traps and Prokopack aspiration indoors and outdoors then identified based on morphological characteristics and screened for fluorescence due to the uranine dye. Molecular species identification was done using polymerase chain reaction and sporozoite infectivity tests by Enzyme-linked immunosorbent assay. Data analysis was performed using R statistical software.

## Results

The predominant malaria vector was *An. funestus sensu lato* (s.l.), which had an overall dye feeding rate of 11.2%. This was followed by *An. gambiae* s.l. at 3.5%. These corresponded to daily feeding rates of 4.8% and 1.2%, respectively. Sporozoite positivity rates were 2.3% (n = 29) in *An. funestus* s.l and 1.0% (n = 19) in *An. gambiae s.l.* Higher dye positivity was observed in male *An. funestus* (OR = 1.22; 95% CI = 1.03,1.46; P = 0.029) and male *An. gambiae* (OR = 2.20; 95% CI = 1.19,4.08; P = 0.015). Comparison of the impact of 2 versus 3 bait stations indicated no difference in feeding rates in either *An. funestus* (OR = 0.83; 95% CI = 0.40; 1.75), P = 0.624) or *An. gambiae* (OR = 1.11; 95% CI = 0.71, 1.71; P = 0.661).

## Conclusions

The results from this study showed that predominant malaria vectors; *Anopheles funestus* led to a daily feeding rate that was presumed to be sufficient to cause a reduction in malaria incidence by ATSBs. There was no significant difference detected between deploying two or three bait stations per structure. The study provided important information utilized in the subsequent deployment of ATSBs in epidemiological trials.

## Introduction

Malaria infected an estimated 263 million people with 597,000 deaths globally in 2023 with the greatest burden occurring in Sub-Saharan Africa [1]. Vector control is an essential component of malaria prevention with the primary tools recommended for use by WHO—long lasting insecticidal nets (LLINs) and indoor residual spraying (IRS)—targeting indoor biting mosquitoes that are largely active when people are in bed or which spend time resting on walls inside the house [2]. While these tools

have been highly effective in reducing malaria transmission and disease, there remain significant gaps in protection where humans can still be bitten and potentially infected. These gaps include new evidence indicating a shift to outdoor feeding and later biting times by mosquitoes that are outside those of typical LLIN usage [3]. Although human landing catch studies indicate the bulk of transmission largely occurs indoors, malaria elimination cannot be achieved without addressing sources of residual transmission. To address these gaps, there is a need for vector control tools that target mosquitoes that feed at times when people are not under nets and mosquitoes that may rest outdoors.

Attractive targeted sugar baits (ATSBs) are a candidate vector control tool that takes advantage of mosquito sugar feeding behaviour [4]. They contain a sugar solution laced with an oral toxin and attractants, designed to lure both male and female mosquitoes to feed and ingest the toxicant, thereby reducing local vector populations. Sugar foraging is an important aspect in the survival of all mosquito species [5–7] and the availability of nectar sources in the local environment is a key regulator of mosquito population dynamics and vectorial capacity [7–9]. Both male and female mosquitoes require sugar for energetic needs, feeding frequently throughout their adult life, and thus could be targeted by ATSBs at various contact points of their lifespan [10]. Small scale studies with entomological outcomes highlighted the promise of ATSBs for the prevention of malaria. The first field trials were conducted in Israel against *Anopheles claviger* in underground cisterns [11] and against *An. sergentii* and *An. caspius* in a desert oasis [12]. Subsequently, a small-scale trial was done in Mali, where vegetation was sprayed with a sugar solution containing boric acid, resulting in an estimated 90% reduction in mosquito densities [13]. More recently, a larger trial using a prototype of a commercial bait station resulted in a significant reduction in mosquitoes older than 3 gonotrophic cycles [14]. However, all previous ATSB trials were done in areas with semi-arid climates and/or extensive dry seasons with limited availability of natural sugar sources. These ecological settings may be the optimal places to implement ATSBs as they may have limited flowering vegetation serving as alternative sugar sources compared to other areas with year-round rainfall and greater abundance of natural sugar sources. To investigate whether bait stations have the potential for use in other ecological settings of Africa, it is necessary to evaluate them in areas with a higher abundance of flowering vegetation. Due to the novelty of the ATSB paradigm in this region, it was paramount to conduct this precursor study using attractive sugar baits (ASBs) to predict the performance of ATSB.

This study evaluated whether mosquitoes would feed on ASBs deployed on the outside walls of houses and other structures in areas of high availability of natural plants that could serve as alternative sources of sugar. The ASBs were similar in design to the prototype ATSB but lacked the active ingredient to kill mosquitoes. The aim of this study was to quantify the dye positivity of malaria vectors that fed on ASBs in western Kenya to estimate the daily feeding rate and to determine whether it exceeded the modelled threshold that would support an epidemiological trial of ATSBs in this region. Additionally, the study sought to determine whether the deployment of three ASB stations per building structure would lead to a significant increase in the dye positivity when compared to two ASB stations per structure for local vectors. The outcome of this investigation directly informed the design and implementation of a subsequent epidemiological trial aimed at evaluating the efficacy of ATSBs for malaria control [15].

## Methods

### Study site

The study was conducted in 12 villages (clusters) in Rarieda Sub-County, Siaya County, western Kenya. The field site is located west of Kisumu town and north of Lake Victoria (S1 Fig). Residents live in family compounds that consist of 1 or more house structures surrounded by farmland. Due to the non-contiguous distribution of compounds in western Kenya, a 'fried-egg' design was adopted, in which each selected village (the 'egg white') incorporated a ≥ 0.2 km buffer from its edge, surrounding a central 'core' area (the 'yolk') where sampling was conducted. This approach minimized potential spill over effects from neighbouring untreated areas. Sampling did not take place in the buffer areas even though ASBs were deployed there. The primary malaria vectors in the study area are *Anopheles funestus, Anopheles arabiensis* and *Anopheles gambiae s.s.* [16,17]. Malaria transmission is year-round, with seasonal peaks during the long rains (March-June)

and the short rains (October-December). In 2020, malaria prevalence in the Lake endemic region of western Kenya was estimated at 19% among the general population [18]. The impact of rainfall on the mosquito feeding rate on ASBs was assessed using weather data including rainfall information collected over the course of the study period from a meteorological station located within the Asembo study area and weekly precipitation averages were calculated.

## Attractive Sugar Baits (ASBs)

The ASBs evaluated in this study was developed by Westham Company (Hod-Hasharon, Israel). It is a prototype product lacking the toxic ingredient incorporated into ATSBs. They were constructed with a white, rectangular plastic frame as the back layer, a permeable black membrane at the front and reservoirs impregnated with a chemical attractant containing a sugar-based feeding stimulant and a fluorescent dye (uranine) for the visualization of mosquitoes that successfully feed on ASBs. Mosquitoes can probe and feed through the membrane to obtain a dye-containing sugar meal. There were two versions of ASB tested during this field study; V1.1.1 was deployed initially and replaced by ASB version V1.2.1 at the crossover point of the study. The switch was necessitated due to a change in the ATSB version that would be utilised in the subsequent epidemiological trials across three countries; Kenya, Mali and Zambia [15]. Version 1.2.1 included minor adaptations to enable higher throughput during manufacture. The ASB version V1.2.1 was used for the remainder of the study.

## Community sensitization and deployment of ASBs

Community sensitization meetings were conducted prior to the field activities. Local administrators, members of the Siaya County Health Management Team, community health workers and area residents were informed about the study. Consent was sought from the household heads to participate in the study. Eligible structures for ASB station deployment were those with a roof over-hang, at least 3 complete walls of heights greater than 1.6 meters from the ground level; structures that did not meet the criteria were excluded from the study. Each structure was mapped, its geolocation recorded, and a corresponding structure code assigned to each. Any new eligible structures built after the initial mapping were assigned a structure ID when they were identified by the study team.

Two or three bait stations, depending on the cluster were mounted approximately 1.8 m from the ground using nails and strings on all eligible structures in consenting compounds. In the two-ASB arm, the bait stations were placed on opposite walls of the structure, while in the three-ASB arm, one device was placed on each of any three walls of the structure. The number of ASB stations deployed in each household was recorded by scanning the serial number on the ASBs during deployment. A total of 11,024 ASB stations were deployed on 4,314 structures during the pre-crossover period, and 10,942 ASBs deployed on 4,434 structures in the post-crossover period (S1 Table).

## Density of ASBs

The spatial density of installed ASB stations and of eligible structures and the weighted spatial density per hectare of installed ASB stations and of eligible structures, were calculated using procedures adapted from Ottensmann [19,20]. In brief, North–South and East–West gridlines were superimposed upon a map of each cluster, with the parallel lines separated by 100 m, dividing the area into 1-hectare squares. Density refers to the number of eligible structures (or ASB stations installed) per hectare within the boundaries of each cluster, while the weighted density refers to the average density of eligible structures (or ASB stations installed) weighted by the number of eligible structures per grid square with at least one eligible structure.

## Study design

An initial cross-over design was employed to evaluate if the deployment of 2 ASBs per structure versus 3 ASBs per structure would lead to a significant difference in the dye positivity rate of vectors in 10 clusters. Two control clusters were

included to monitor for false positives. For the clusters that received ASBs, a changeover was conducted where the initial allocation of 2 versus 3 ASBs per structure was switched over to allow for a contrast of the deployment density in all intervention clusters. The crossover design evaluation of 2 vs 3 ASB stations began on 14 May 2021 with the deployment of V1.1.1 for a 2-month pre crossover evaluation period beginning 23 May 2021. The changeover to the V1.2.1 and swapping the clusters receiving either 2 or 3 ASBs was carried out between 26 July 2021and 1 August 2021. The post crossover period continued for a further 2 months with mosquito collections finishing on 26 September. Sampling for the V1.2.1 evaluation continued with the existing ASB deployment for an additional 4 months. This led to two separate but overlapping evaluation periods, the crossover assessment of V1.1.1 and V1.2.1 to evaluate 2 vs 3 stations (May to September 2021) and the evaluation of V1.2.1 (August 2021 to January 2022). The two control clusters were not monitored beyond the crossover trial period.

## Mosquito sampling

Mosquito sampling was done every month in 10 randomly selected compounds from the core area of each cluster. Five different compounds were randomized for sampling per cluster every week and mosquito collections were done twice per month on a rotational basis. Compounds located within the 0.2 km buffer zone at the periphery of each cluster received ASBs but were not eligible for mosquito sampling. Mosquito collections were conducted inside and outside sleeping structures using either ultraviolet CDC light traps (UVLTs) (Model 912, John W. Hock Company, Gainesville, Florida, USA) or Prokopack aspiration (Model 1419, John W. Hock Company, Gainesville, Florida, USA) [21] targeting pots and other potential mosquito resting places around the compound. Mosquitoes were collected from up to 10 compounds per cluster per month using UV light traps and from an additional 10 households close to those of UVLTs using Prokopack aspiration. Indoors, the light traps were set 1.5 m above the ground at the foot of an occupied bed net. Outdoors, the traps were placed approximately 5 meters from the structure with an indoor trap but were not baited with either a human or an animal or with carbon dioxide. They were suspended from unoccupied buildings or trees approximately 1.5 m above the ground and shielded from rain. All light traps were set up at 5:00 pm and collected at 07:00 am the following morning. Prokopack aspirations were done indoors and outdoors in one house structure in each of the 10 selected compounds from 07.00 to 11.00 am. The sampled mosquitoes from each collection were transferred to labelled paper cups per collection, separating the outdoor and indoor catches. Each collection cup, from the UVLT and aspiration, was labelled with a unique collection and house code generated from a tablet for every sampled house structure and then sent to the field laboratory in Lwak, Asembo, for further processing.

## Mosquito sorting and feeding assessment

Upon arrival at the laboratory, the sampled insects were killed using either chloroform fumes or by freezing at −20˚C. The insects were sorted and counted after which non-*Anopheles* mosquitoes were discarded. The *Anopheles* mosquitoes were separated by species, sex, and the abdominal status (blood fed, non-blood fed, or gravid) for females and numbers collected per trap recorded. Mosquitoes were identified by morphological features using taxonomic keys [22] to differentiate between *An. funestus s.l.*, *An. gambiae s.l.*, and other secondary malaria vectors. The mosquitoes were assessed for the presence or absence of uranine dye using a fluorescence microscope (Leica MZ10F FLUO III™, Leica Microsystems, Morrisville, USA). Samples were frozen at -20ºC until they were read with reading being completed for samples within 7 days of collection. Verified positive and negative controls were used for comparisons and confirmation of results. Each mosquito sample was read independently by two dye readers who were blinded to the source cluster. Ongoing quality assurance checks were conducted using random verified positive and negative controls to ensure accuracy of the readers. If the readers produced discordant results, a third reader would conduct a final read for definitive assignment.

A subset of 40 *Anopheles funestus s.l.* and *Anopheles gambiae s.l.* per cluster per month was selected at random from samples collected and transported to the main entomology laboratory at KEMRI-CGHR in Kisumu for molecular analysis.

Sometimes this target was not attained in some clusters and when that happened, all the samples from those clusters were utilized for molecular processing. Mosquitoes were divided into head and thorax for sporozoite analysis, and abdomens, legs and wings for species identification using polymerase chain reaction (PCR) as per previous protocols [23]. Sporozoite infection rates were determined by enzyme-linked immunosorbent assay (ELISA) using standard protocols [24,25].

### Monitoring of the ASB stations

ASBs deployed to households were monitored every two weeks to check for any missing, visible leaks, mould growth and damage. This survey was completed on tablets, recording information on the station's location and whether they were damaged. During the monitoring visits, bait stations were replaced if they met the following criteria: they were torn, soiled (covered with paint or mud), mouldy, holed, bait was leaking, or bait was depleted in five or more cells. The number of lost bait stations was recorded and the reasons for bait station replacement were assessed and summarized.

### Conversion of dye-positivity to daily feeding rates

Daily feeding rates were estimated from the proportion of uranine-positive mosquitoes using a polynomial model ($F = au^2 + bu$) developed at Imperial College London, where $F$ is the proportion positive and $u$ the daily feeding rate. Coefficients ($a$ and $b$) were adjusted for different feeding rate ranges (≤10% or 10–50% per day), assuming a mean dye persistence of 4.5 ± 0.5 days. The feeding rate was derived as $u = [\sqrt{(b^2 + 4aF)} - b]/(2a)$, with bounds calculated under assumptions of 9.6% natural daily mortality, no excess mortality from uranine, and 9% additional mortality due to ITNs and IRS. Variation in mosquito population size had little effect on the estimates, justifying the assumption of a constant population.

### Data analysis

For the cross-over trial period, a comparison of dye positivity between clusters with two versus three ASBs was carried out for the two major *Anopheles* mosquito species while adjusting for confounding factors including period effects, with Bonferroni correction applied to account for multiple testing (two species comparisons). The analysis used a generalised linear mixed model (GLMM) with a binomial distribution and random effects for cluster, household, collection date, and cluster-day interactions to account for the hierarchical and repeated measures structure of the data. To characterize heterogeneity in ATSB feeding patterns during the extended monitoring period (post crossover), we conducted further exploratory comparisons which included comparisons between *An. funestus s.l.* and *An. gambiae s.l.,* and species-specific comparisons of 2 vs 3 stations, indoor versus outdoor mosquito collections, and males versus females. These comparisons were conducted using a GLMM with a binomial distribution of the proportion dye positive and a random effect for clusters to generate an odds ratio of the effect. To estimate the daily feeding rates, the cluster level data were converted to daily feeding rate using the approach outlined in [26]. Briefly, the model accounts for dye decay, natural mortality and other interventions in the area to convert the total dye positivity to an estimated daily feeding rate that is assumed equivalent to the reduction in daily survival that could be attributed to ATSBs. The effects of bait station density and rainfall on mosquito feeding rates were evaluated using the GLMM framework. Bait station density was defined as the number of ASBs deployed per cluster, standardized by cluster size, and included as a continuous predictor. Rainfall data were averaged on a weekly basis over the study period, and aligned with the corresponding weeks of mosquito collection. Rainfall was treated as a continuous covariate. In the models, the dependent variable was the proportion of mosquitoes positive for uranine dye, with bait station density and rainfall as fixed effects and cluster as a random effect. All models were generated using the lme4 package in R version 4.4.1.

### Ethical consideration

This study was reviewed and approved by the Kenya Medical Research Institute Scientific and Ethical Review Unit (KEMRI SERU; Approval No. 3613). Additional approvals were obtained from the Liverpool School of Tropical Medicine

Research Ethics Committee (Approval No. LSTM 18–015) and the U.S. Centers for Disease Control and Prevention (CDC; Approval No. 7112) under a reliance agreement with KEMRI SERU. Voluntary, informed consent was obtained from the head of each household enrolled in the study to receive sugar baits containing a marker dye. For mosquito collections, oral consent was sought from households on the day of collection before placing UV light traps. All consent procedures were conducted in the local languages (DhoLuo and Kiswahili) and back-translated into English to ensure accuracy.

## Results

### *Anopheles* species abundance

A total of 34,457 *Anopheles* mosquitoes comprising of 21,337 *An. funestus s.l.* (15,173 females) and 5,145 *An. gambiae s.l.* (4,145 females) were collected from the 12 study clusters using indoor and outdoor UV light traps and aspiration across the entire sampling period. Other *Anopheles* mosquitoes collected included: 7,326 *An. coustani*, 348 *An. maculipalpis*, 235 *An. pharoensis*, 49 *An. rufipes* and 17 *An. squamosus*.

Of the 1,327 *An. funestus s.l.* samples analysed by PCR, 82% (n = 1086) were confirmed to be *An. funestus s.s.* and 6.3% (n = 83) *An. leesoni.* The remaining 12% (n = 158) did not amplify. Of 1,964 *An. gambiae s.l.* that were assayed, 68% (n = 1,342) were confirmed to be *An. arabiensis* and 21% (n = 407) *An. gambiae s.s.* The remaining 11% (n = 215) did not amplify. A sample of the two predominant species *An. funestus s.l.* (n = 1,273)*,* and *An. gambiae s.l.* (n = 1,894) were analysed for *Plasmodium falciparum* sporozoite infection. *Anopheles funestus s.l.* samples had a sporozoite positivity rate of 2.3% (n = 29) while *An. gambiae s.l.* samples recorded a sporozoite positivity rate of 1.0% (n = 19).

### Bait station density and ASB monitoring data

Overall, the average number of ASB stations deployed per structure was 2.36 and 2.23 during the pre-crossover and post-crossover periods respectively. In the arm with 2 ASBs per structure, an average of 1.85 stations were deployed at pre-crossover and 1.79 post-crossover whereas in the 3 ASB arm, an average of 2.79 and 2.74 stations were deployed per structure during the pre- and post-crossover periods respectively. The average number was slightly below the target due to a small number of households that did not consent to have ASBs deployed around their homes. Bait station density deployed by cluster is summarized in S2 Table.

For ASB version 1.1.1 in the pre-crossover period, of the 11,024 that were initially deployed, 89 (0.81%) were replaced for meeting one or more of the failure criteria of being torn, soiled, holed, leaking, mouldy or had bait depleted in five or more cells. For the version 1.2.1 deployed in the second crossover period and then monitored for an additional 4 months, 71 (0.65%) of the 10,942 deployed met the replacement criteria for the comparable trial period that V.1.1.1 was assessed. For the entire cross over evaluation period (8 months) a total of 1113 ASBs were replaced. The most common reasons for ASB replacement were physical damage 601 (54.0%) followed by the presence of mould 394 (35.4%) and leakage 7 (0.63%). Reasons for the remaining 111 (9.97%) included being removed by household members, the structures being demolished, and residents not knowing the whereabouts of the ASB.

### Rates of dye positivity in primary malaria vectors

During the entire study period all the 34,457 *Anopheles* mosquitoes collected were observed for dye positivity. The overall dye positivity rate in *Anopheles* mosquitoes was 8.6% (2,951) with the highest rates observed in *An. funestus* mosquitoes representing 83.4% (2,477) of dye positive *Anopheles* followed by *An. gambiae* with 6.2% (183) of dye positive *Anopheles*. Most 33,540 (97%) of the dye read results were confirmed by two independent readers. Only 2.7% (917) of the collected *Anopheles* mosquitoes including 616 females and 301 males required a third reader for confirmation.

For the cross-over trial period (May – Sept), a total of 15,466 *An. funestus* comprised of 10,857 females and 4,609 males and 3,163 *An. gambiae* comprised of 2,603 females and 560 males were assessed for dye positivity. Dye positivity

rates for *An. funestus* were 11.2% (95% CI 7.7–15.7%) for clusters that received 2 ASBs and 12.0% (95% CI 8.3–16.7%) for those that received 3 ASBs. For *An. gambiae s.l.,* dye positivity rates were 3.8% (95% CI 2.2–6.4%) in clusters that received 2 ASBs and 4.5% (95% CI 2.5–7.8%) for those that received 3 ASBs.

After adjusting for period effects, there was no significant difference in proportion dye positive between clusters with two or three ASBs for either *An. funestus* (OR = 1.09; 95% CI = 0.71, 1.98; P = 0.661; Bonferroni-adjusted P = >0.999) or *An. gambiae* (OR = 1.00; 95% CI = 0.52, 1.93; P = 0.996; Bonferroni-adjusted P = >0.999) (S2 Fig). A significant period effect was observed for *An. gambiae* (P = 0.032), but not for *An. funestus* (P = 0.093), indicating that the crossover design successfully controlled for temporal confounding in the primary comparison.

During the extended evaluation period (Aug 2021– Jan 2022), a total of 8,408 *An. funestus* and 2,536 *An. gambiae* were assessed for dye positivity. The *An. funestus* included 5,896 females and 2,512 males while *An. gambiae* included 1,971 females and 565 males. When the rates of mosquito dye positivity were compared across the ten treatment clusters, *An. gambiae* had a lower positivity rate compared to *An. funestus* in most of the clusters (OR = 0.19; 95% CI = 0.14, 0.27; P < 0.001) (S3 Fig A). When dye positivity was assessed by collection location (indoors or outdoors), no significant differences in dye positivity rates were observed among *An. funestus* (OR = 1.40; 95% CI = 0.99, 1.97; P = 0.056) (S3B Fig) while significantly higher dye positivity rates were recorded in *An. gambiae* collected indoors compared to outdoors (OR = 2.97; 95% CI = 0.97, 9.05; P = 0.031) (S3C Fig). When comparisons of dye positivity were done by species and sex, there were higher dye positivity rates in males compared to females for both *An. funestus* (OR = 1.22; 95% CI = 1.03, 1.46; P = 0.029) and *An. gambiae* (OR = 2.20; 95% CI = 1.19, 4.08; P = 0.015) (S3D and S3E Fig). Dye positivity was higher in *An. funestus* than *An. gambiae* during every month of the post cross-over and extended monitoring period (S3F Fig).

### ASB daily feeding rates from dye positivity

When the proportions of mosquitoes collected that were positive for uranine dye were converted to an estimated daily feeding rate as previously outlined in a similar study conducted in Zambia [26], the overall daily feeding rate on ASBs was estimated at 4.8% (95% CI 2.4–9.4%) for *An. funestus* and 1.2% (95% CI 0.51−2.8%) for *An. gambiae* (S4 Fig). The model accounts for the impact of natural mortality, caused by other control tools and the longevity of the dye, since dye remains detectable several days after initial feeding. These results were compared to the modelled target daily feeding rate needed to achieve a 30% reduction in clinical incidence after 1 year of deployment [27], and were found to be above the target.

### Relationship of bait station density and rainfall to dye positivity

There was significant variability in the rainfall throughout the study period with a mean weekly rainfall of 0.5 mm in the pre-cross over period, 41.9 mm in post-cross over period, and 26.3 mm in the extended observation period (S5 Fig). For *An. funestus*, weekly precipitation (cm) demonstrated a significant positive association (OR = 1.02; 95% CI = 1.01–1.02; P = <0.001) on dye positivity, while cluster bait station density showed no statistically significant effect (OR = 1.01; 95% CI = 0.99–1.03; P = 0.285). Conversely, for *An. gambiae*, cluster bait density had a significant negative effect (OR = 0.91; 95% CI = 0.85–0.98; P = 0.016) on dye positivity while precipitation did not have a statistically significant impact (OR = 1.01; 95% CI = 0.99–1.04; P = 0.286). Assessed together, the precipitation effect remained consistent and significant on dye positivity (OR = 1.02; 95% CI = 1.01–1.03; P = <0.001), while bait station density continued to show no significant impact (OR = 0.99; 95% CI = 0.96–1.01; P = 0.246) for *An. funestus*. For *An. gambiae*, both factors achieved statistical significance with bait station density exhibiting a stronger negative effect (OR = 0.88; 95% CI = 0.81–0.95; P = <0.001) while precipitation demonstrated a positive association (OR = 1.04; 95% CI = 1.01–1.07; P = 0.002).

### Dye positivity in control clusters

A total of 91 out of the 2,660 (3.4%) primary *Anopheles spp.* sampled from the control clusters during the entire trial were found to be dye positive. Most were *An. funestus* (86 of 91) and 5 were *An. gambiae*. The minimum distance was

calculated to the nearest structure that had an ASB station present. This was 1644m, 1387m and 1061m for *An. funestus* females, *An. funestus* males and *An. gambiae* females, respectively.

## Discussion

This study represents a field-based study of attractive sugar baits in Kenya and provided important information used in the decision to proceed with the pattern of deployment of ATSBs in subsequent epidemiological trials. The lack of difference in mosquito dye positivity rates in villages that received two versus three bait stations in this and other trials informed the decision to use two bait stations per structure in subsequent Phase III epidemiological trials of ATSBs that were initiated in Mali, Zambia and Kenya [15]. These results confirm previous findings from Mali and Zambia where the percentage of marked mosquitoes trapped when using two or three bait stations were not statistically different between the two arms [26,28]. This study also demonstrated that the males and females of *An. funestus* and *An. gambiae* populations collected were dye positive which suggests that ATSBs may potentially impact local mosquito populations when used in addition to existing vector control interventions. Although only female *Anopheles* mosquitoes transmit malaria, high male feeding rates still play a critical role in supporting overall malaria vector control. Males make up a substantial portion of the mosquito population, and their mortality contributes to reducing population size and mating potential, thereby indirectly limiting transmission [13,29]. The higher male feeding observed in our study may also serve as an indicator of bait attractiveness. However, maximizing female exposure remains critical since epidemiological impact depends primarily on reducing female survival and vectorial capacity [30].

The findings from this study demonstrated higher daily feeding rates among *An. funestus* than *An. gambiae* further confirming the findings of a previous study from Zambia [26]. While the observed feeding rates seemed to surpass the estimated threshold required to demonstrate an epidemiological impact if the intervention had a toxic insecticide incorporated, subsequent epidemiological studies assessing the efficacy of ATSBs conducted in Zambia and Kenya did not find any clear impact on malaria incidence or prevalence [31,32]. The reason for the discrepancy between the expected and observed impact of ATSBs is unclear but may be explained by several factors, including inaccuracies in the modelled estimate of the impact of the ATSBs, which assumed a constant daily feeding rate across the entire population of mosquitoes, or biases in the collection and/or scoring of mosquitoes for dye positivity due to spatial heterogeneities or the proximity of the of traps to the ASBs, which could have overestimate the daily feeding rate. Despite the unexpected outcome of the successive main trials, this study remains important in characterizing the mosquito daily feeding rates in future ATSB validation studies and further modelling work should be conducted to better understand the relationship between daily feeding rates and epidemiological impact.

The mosquito daily feeding rates found in Kenya were generally lower than those found in previous studies in Zambia [26] and Mali [14,28]. This may be attributed to differences in ecological zones, times when the studies were conducted, and differences in mosquito species found in the research sites. Although a semi-field study conducted in Kenya demonstrated that ATSBs were more attractive to local *Anopheles* mosquitoes compared to the most attractive natural sugar source tested [33], differences in environmental conditions and variability in availability of natural sugar sources under natural settings may have reduced or masked the attractiveness of the ASBs in western Kenya. It was also observed that mosquitoes were more likely to feed on the ASBs during wetter periods. Though somewhat counterintuitive, the ASBs tended to absorb water and swell during wetter, more humid periods, often resulting in the contents of the ASB seeping out onto the surface. During dry periods, mosquitoes may have had more difficulty in accessing the contents of the ASBs but, despite the lack of rainfall, there was still plenty of flowering vegetation for mosquitoes to feed upon.

There were a small number of dye positive mosquitoes that were collected in the control clusters. This calls for studies to understand the dispersal distance of malaria vectors in this region since it is relevant in designing trials of ATSBs as well as other vector control interventions. Previous work in this area revealed community impacts of ITN-based vector control interventions up to 600m from the intervention cluster [34,35] suggesting limited movement of mosquitoes between

villages. In a study in Tanzania, the average dispersal distance was reported to be 510 m for *An. funestus* and 654 m for *An. arabiensis* [36], although over 20% of *An. funestus* were collected 860m from the release point. These findings suggest that buffer zones of up to 1000m or more should be included in trials of vector control products to minimize the risk of contamination.

The study had a number of limitations, including small cluster sizes with buffer distances of only 200m. The finding of dye positive mosquitoes in the control villages that were over 1 km away from ASB villages may mean that feeding rates were underestimated as mosquitoes may migrate into the ASB clusters. Conversely, it is possible that the dye positive mosquitoes that were collected from the control clusters might have resulted from autofluorescence suggesting the feeding rates in the ASB villages may have been overestimated. However, the dye readings were the consensus of at least two readers who were blinded to the treatment allocation. Furthermore, quality assurance testing was done by regularly testing the readers with mosquitoes that were known to be unfed or to have fed on the ASB solution to ensure the accuracy of their determinations. The use of different ASB versions during the trial potentially confounded comparisons of the two versus three ASBs on dye positivity rates and impacted the ability to account for other time-varying covariates such as rainfall in the cross over study. Indeed, variation in bait attractiveness between versions has been documented in semi-field comparisons which reinforces the possibility that version changes introduced systematic bias in comparing arms [33].

## Conclusion

The deployment of ASB stations on the outside walls of structures in villages in western Kenya resulted in an estimated daily feeding rate in the primary malaria vectors, *An. funestus* and *An. gambiae*, that may result in at least a 30% reduction in the disease' incidence. These findings met the study objective of quantifying mosquito feeding rates and demonstrated that the observed rates exceeded the modelled threshold required to justify an epidemiological evaluation of ATSBs in this region. However, subsequent epidemiological studies assessing the efficacy of ATSBs conducted in Zambia and Kenya did not find any clear impact on malaria incidence or prevalence. No difference in dye positivity rates were observed in villages with two versus three bait stations suggesting that deployment of two ATSBs per structure would be adequate to reduce malaria transmission in western Kenya and an additional bait station per structure would provide no additional benefit. This study offered valuable insights that were utilized in the later implementation of the larger epidemiological trials and the exploratory analysis in the extended period provided some interest areas for future studies.

## Supporting information

**S1 Fig. Map of study villages in Siaya County, western Kenya.**
(PNG)

**S2 Fig. Comparison of the impact of 2 versus 3 bait stations on *An. funestus* and *An. gambiae* dye positivity.** The upper and lower bounderies of the error bars represent the 95% confidence interval and 'NS' if not significant.
(DOCX)

**S3 Fig. Comparison of percentage dye positivity of *Anopheles* mosquitoes by; cluster (A), indoor and outdoor sampling locations for *An. funestus* (B) and *An. gambiae* (C), sex for *An. funestus* (D) and *An. gambiae* (E) and collection month (F).** P-value significance is indicated by '*' if ≤ 0.05, '***' if ≤ 0.001 with the upper and lower bounderies indicating the 95% confidence interval and 'NS' if not significant.
(DOCX)

**S4 Fig. Estimation of daily feeding rate (DFR) generated from the study derived dye positivity (red diamond) with the associated 95% confidence interval overlaid on the estimated percentage reduction in malaria incidence in one year in western Kenya for a given *An. funestus* and *An. gambiae* DFR.** The middle dashed black line displays the

estimated DFR required to achieve a 30% reduction with the upper and lower bounderies indicating the 95% confidence interval of this estimate.
(DOCX)

**S5 Fig. Weekly precipitation (mm) for Siaya County (January 2021–January 2022).** The crossover trial period is high-lighted by "Pre" (pink shading), "Post" (blue shading), and "Extended" (green shading). Horizontal coloured dashed lines indicate the mean weekly rainfall for the associated colour period.
(DOCX)

**S1 Table. Number of ASB stations and structure allocations during pre and post crossover period.**
(DOCX)

**S2 Table. Bait station density by cluster.**
(DOCX)

## Acknowledgments

Our sincere gratitude goes to the communities where the study was conducted, and especially the household owners who allowed product deployment and granted access to their houses for mosquito collection and study activities. Our thanks also go to the entomology field and laboratory teams for their dedicated efforts and quick turn-around time in sample processing. We express our appreciation to Siaya County for the technical and logistical support.

Westham Co. (Hod Hasharon, Israel) provided the ASB stations used in this study.

## Author contributions

**Conceptualization:** Seline Omondi, Martin J. Donnelly, Angela F Harris, Julian Entwistle, Feiko O Ter Kuile, John E. Gimnig, Eric Ochomo.

**Data curation:** Daniel P. McDermott, Vincent Moshi, Prashanth Selvaraj, Keith Fraser, Lazaro Mwandigha, Bernard Abong'o, John E. Gimnig, Eric Ochomo.

**Formal analysis:** Jackline Jeruto Kosgei, Daniel P. McDermott, Vincent Moshi, Prashanth Selvaraj, Keith Fraser, Lazaro Mwandigha, Bernard Abong'o, John E. Gimnig, Eric Ochomo.

**Funding acquisition:** Angela F Harris, Julian Entwistle, John E. Gimnig, Eric Ochomo.

**Investigation:** Martin J. Donnelly, Collins Ouma, Julian Entwistle, Feiko O Ter Kuile, John E. Gimnig, Eric Ochomo.

**Methodology:** Jackline Jeruto Kosgei, Daniel P. McDermott, Seline Omondi, Martin J. Donnelly, Prashanth Selvaraj, Angela F Harris, Julian Entwistle, Feiko O Ter Kuile, Keith Fraser, Bernard Abong'o, John E. Gimnig, Eric Ochomo.

**Resources:** John E. Gimnig, Eric Ochomo.

**Supervision:** Martin J. Donnelly, Collins Ouma, Bernard Abong'o, John E. Gimnig, Eric Ochomo.

**Validation:** Lazaro Mwandigha, John E. Gimnig, Eric Ochomo.

**Visualization:** Jackline Jeruto Kosgei, Daniel P. McDermott, John E. Gimnig, Eric Ochomo.

**Writing – original draft:** Jackline Jeruto Kosgei.

**Writing – review & editing:** Jackline Jeruto Kosgei, Daniel P. McDermott, Seline Omondi, Vincent Moshi, Martin J. Donnelly, Collins Ouma, Prashanth Selvaraj, Angela F. Harris, Julian Entwistle, Feiko O. Ter Kuile, Keith Fraser, Lazaro Mwandigha, Bernard Abong'o, John E. Gimnig, Eric Ochomo.

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
