## [Decision Letter · Decision Letter 0]

14 Aug 2025

Dear Dr. Kosgei,

Thank you for submitting your manuscript to PLOS ONE. After careful consideration, we feel that it has merit but does not fully meet PLOS ONE’s publication criteria as it currently stands. Therefore, we invite you to submit a revised version of the manuscript that addresses the points raised during the review process.

Please pay particular attention to the comments from Reviewer 3 as they raise important issues regarding the clarity of the manuscript and its adherence to the stated goals of the analysis. Please also address all reviewer's comments and suggestions in any subsequent drafts of the manuscript, as all had relatively minor but important recommendations regarding grammar.

We look forward to receiving your revised manuscript.

Kind regards,

James Colborn

Academic Editor

PLOS ONE

Journal Requirements:

[This study was funded by IVCC through support from the Bill and Melinda Gates Foundation (INV-

007509), the Swiss Agency for Development and Cooperation (SDC) (Grant: 81067480) and UK Aid (Grant: 30041-105). The findings and conclusions contained within are those of the authors and do not necessarily reflect positions or policies of IVCC, the Bill & Melinda Gates Foundation, SDC or UK Aid.].

4. Please expand the acronym “IVCC” (as indicated in your financial disclosure) so that it states the name of your funders in full.

[We also acknowledge IVCC for funding the study through support from the Bill and Melinda Gates Foundation, the Swiss Agency for Development and Cooperation (SDC) and UK Aid.]

[This study was funded by IVCC through support from the Bill and Melinda Gates Foundation (INV-

007509), the Swiss Agency for Development and Cooperation (SDC) (Grant: 81067480) and UK Aid (Grant: 30041-105). The findings and conclusions contained within are those of the authors and do not necessarily reflect positions or policies of IVCC, the Bill & Melinda Gates Foundation, SDC or UK Aid.]

7. Please amend the manuscript submission data (via Edit Submission) to include author Jackline Kosgei.

8. Please amend your authorship list in your manuscript file to include author Jackline Jeruto Kosgei.

9. Please include your full ethics statement in the ‘Methods’ section of your manuscript file. In your statement, please include the full name of the IRB or ethics committee who approved or waived your study, as well as whether or not you obtained informed written or verbal consent. If consent was waived for your study, please include this information in your statement as well.

10. Please include a separate caption for each figure in your manuscript.

11. We note that Supplemental Figure 1 in your submission contain map images which may be copyrighted. All PLOS content is published under the Creative Commons Attribution License (CC BY 4.0), which means that the manuscript, images, and Supporting Information files will be freely available online, and any third party is permitted to access, download, copy, distribute, and use these materials in any way, even commercially, with proper attribution. For these reasons, we cannot publish previously copyrighted maps or satellite images created using proprietary data, such as Google software (Google Maps, Street View, and Earth). For more information, see our copyright guidelines: http://journals.plos.org/plosone/s/licenses-and-copyright.

1. You may seek permission from the original copyright holder of Supplemental Figure 1 to publish the content specifically under the CC BY 4.0 license. 

12. We notice that your supplementary figures are uploaded with the file type 'Figure'. Please amend the file type to 'Supporting Information'. Please ensure that each Supporting Information file has a legend listed in the manuscript after the references list.

13. Please include your tables as part of your main manuscript and remove the individual files. Please note that supplementary tables be uploaded as separate "supporting information" files.

14. Please remove your figures from within your manuscript file, leaving only the individual TIFF/EPS image files, uploaded separately. These will be automatically included in the reviewers’ PDF.

15. Please include captions for your Supporting Information files at the end of your manuscript, and update any in-text citations to match accordingly. Please see our Supporting Information guidelines for more information: http://journals.plos.org/plosone/s/supporting-information.

Reviewers' comments:

Reviewer's Responses to Questions

**Comments to the Author**

1. Is the manuscript technically sound, and do the data support the conclusions?

Reviewer #1: Yes

Reviewer #2: Yes

Reviewer #3: Yes

2. Has the statistical analysis been performed appropriately and rigorously?

Reviewer #1: Yes

Reviewer #2: Yes

Reviewer #3: Yes

3. Have the authors made all data underlying the findings in their manuscript fully available?

Reviewer #1: Yes

Reviewer #2: Yes

Reviewer #3: Yes

4. Is the manuscript presented in an intelligible fashion and written in standard English?

Reviewer #1: Yes

Reviewer #2: Yes

Reviewer #3: Yes

Reviewer #1: The manuscript is well-structured, scientifically rigorous, and highly relevant to the field of malaria vector control as it reports a robust field study that evaluates the sugar feeding behavior of malaria vectors using Attractive Sugar Baits (ASBs) in western Kenya. The research fills an important gap in understanding vector behavior in regions with abundant natural sugar sources and informs deployment strategies for ATSBs in subsequent large-scale trials. There are few suggestions made in my reveiw comments for consideration, including suggestions of the inclusion of some important data into the manuscript.

Reviewer #2: There are typos that need to be corrected before publishing,

Progress of malaria vector control has not stalled since 2015. It is ongoing

Instead of using understand use Assess

Where it is statistically insignificant indicate the statistic result

Exophilic malaria vectors are known

Reviewer #3: The authors report on a field validation study of Attractive Sugar Baits (ASBs) in western Kenya using uranine dye to measure mosquito feeding rates. While methodologically sound, the manuscript requires revision to address critical gaps and improve clarity.

1. The study predicted sufficient feeding rates for epidemiological impact, yet subsequent trials showed no efficacy. The authors inadequately explain this fundamental disconnect. The manuscript can benefit from detailed review of the modeling assumptions, potential validation methodology limitations, and specific comparison with epidemiological trial outcomes.

2. The crossover design is missing a key period effect analysis to test for differences in outcomes over time, independent of the treatment. This is particularly important due to the use of different ASB versions, seasonal variation, and time-dependent confounders. The statistical analysis should also be adjusted for multiple comparisons

Abstract

Line 30: rephrase

Line 54-56: Consider breaking it down for better readability, e.g., "The predominant malaria vector was An. funestus sensu lato (s.l.) with an overall dye feeding rate of 11.2%, followed by An. gambiae s.l. at 3.5%. These corresponded to daily feeding rates of 4.8% and 1.2%, respectively."

Introduction

Line 73 replace “falling” with “lies”.

Line 114-115: This sentence is a bit passive rephrase.

Methods

Line 121: consider replacing “non-clustered” with “non-contiguous” for better clarity.

Line 122: Explain the terminology “fried egg” earlier.

Line 136 replace “which” with “that”.

Line 145: rephrase “was maintained” with a more active phrase such as “was used”.

Line 159-161: The sentence is a little clunky please rephrase.

Discussion

Line 425: Consider including in the methods and results the laboratory comparison of the two versions of ASBs.

**Do you want your identity to be public for this peer review?** For information about this choice, including consent withdrawal, please see our Privacy Policy

Reviewer #1: No

Reviewer #2: No

Reviewer #3: No

---

## [Author Response · Author response to Decision Letter 1]

30 Oct 2025

We sincerely thank the reviewers and the Academic Editor for their valuable feedback, which has helped us improve the clarity and quality of our manuscript

---

## [Decision Letter · Decision Letter 1]

6 Nov 2025

Quantification of Anopheles daily sugar feeding rates in Siaya county, western Kenya using Attractive Sugar Baits

PONE-D-25-29347R1

Dear Dr. Kosgei,

We’re pleased to inform you that your manuscript has been judged scientifically suitable for publication and will be formally accepted for publication once it meets all outstanding technical requirements.

Kind regards,

James Colborn

Academic Editor

PLOS ONE

Additional Editor Comments (optional):

Reviewers' comments:

Reviewer's Responses to Questions

**Comments to the Author**

Reviewer #1: All comments have been addressed

Reviewer #3: All comments have been addressed

2. Is the manuscript technically sound, and do the data support the conclusions?

Reviewer #1: Yes

Reviewer #3: Yes

3. Has the statistical analysis been performed appropriately and rigorously?

Reviewer #1: Yes

Reviewer #3: Yes

4. Have the authors made all data underlying the findings in their manuscript fully available?

Reviewer #1: Yes

Reviewer #3: Yes

5. Is the manuscript presented in an intelligible fashion and written in standard English?

Reviewer #1: Yes

Reviewer #3: Yes

Reviewer #1: I am satisfied with the authors’ responses and the revisions made to the manuscript titled “Quantification of Anopheles Daily Sugar Feeding Rates in Siaya County, Western Kenya Using Attractive Sugar Baits” (PONE-D-25-29347R1). I have no further comments. Congratulations to Mrs. Jackline Jeruto Kosgei and co-authors on their excellent work!

Reviewer #3: I have read through the reviewers' response, and I'm confident that the author has adequately addressed the comments raised.

**Do you want your identity to be public for this peer review?** For information about this choice, including consent withdrawal, please see our Privacy Policy

Reviewer #1: No

Reviewer #3: No

---

## [Editor Report · Acceptance letter]

PONE-D-25-29347R1

PLOS ONE

Dear Dr. Kosgei,

I'm pleased to inform you that your manuscript has been deemed suitable for publication in PLOS ONE. Congratulations! Your manuscript is now being handed over to our production team.

Kind regards,

on behalf of

Dr. James Colborn

Academic Editor

PLOS ONE